# lncRNA TRHDE-AS1 Correlated with Genomic Landscape and Clinical Outcome in Glioma

**DOI:** 10.3390/genes14051052

**Published:** 2023-05-08

**Authors:** Jinxuan Xie, Yi Lin, Yajie Li, Aizhong Fang, Xin Li, Songlin Wang, Wenbin Li

**Affiliations:** 1School of Public Health, Capital Medical University, Beijing 100069, China; 2National Institute for Data Science in Health and Medicine, Capital Medical University, Beijing 100069, China; 3Department of Neuro-Oncology, Cancer Center, Beijing Tiantan Hospital, Capital Medical University, Beijing 100070, China; 4Salivary Gland Disease Center and Beijing Key Laboratory of Tooth Regeneration and Function Reconstruction, School of Stomatology, Beijing Laboratory of Oral Health, Capital Medical University, Beijing 100050, China; 5Department of Biochemistry and Molecular Biology, School of Basic Medical Sciences, Capital Medical University, Beijing 100069, China

**Keywords:** glioma, TRHDE-AS1, lncRNA, bioinformatic

## Abstract

The role of lncRNA in cancer development has received more and more attention in research. A variety of lncRNAs are associated with the occurrence and development of glioma. However, the role of TRHDE-AS1 in glioma is still unknown. In this study, we explored the role of TRHDE-AS1 in glioma through bioinformatic methods. We first identified an association between TRHDE-AS1 and tumor prognosis in pan-cancer analysis. Subsequently, the expression levels of TRHDE-AS1 in various clinical types of glioma were compared, and significant differences were found in pathological classification, WHO classification, molecular classification, IDH mutation, and age stratification. We analyzed the genes co-expressed with TRHDE-AS1 in glioma. In the functional analysis of TRHDE-AS1, we found that TRHDE-AS1 may be involved in the regulation of synapse-related functions. In glioma cancer driver gene correlation analysis, it was also found that TRHDE-AS1 was significantly correlated with the expression levels of multiple driver genes such as TP53, BRAF, and IDH1. By comparing the mutant profiles of the high and low TRHDE-AS1 groups, we also found that there may be differences in TP53 and CIC gene mutations in low-grade gliomas. Subsequent correlation analysis between TRHDE-AS1 and glioma immune microenvironment showed that the expression level of TRHDE-AS1 was correlated with a variety of immune cells. Therefore, we believe that TRHDE-AS1 is involved in the occurrence and development of glioma and has the ability to predict the prognosis of glioma as a biomarker of glioma.

## 1. Introduction

Glioma is the most common primary intracranial neuroepithelial tumor, and it is the most malignant and aggressive tumor in the brain [1], which originates from astrocytes or astrocyte precursor cells. Globally, the incidence rate of glioma is 4.67–5.73/100,000 [1]. Through many pieces of research within the last decades, the explicit mechanism of tumor origin and growth still remains unknown. Nowadays, the main treatment is surgery based on the traditional histology type, and the prognosis is poor. Traditional classification based on histology has its limitations, and so many researchers are demonstrating the significance of molecular biomarkers [2]. The 2016 WHO classification [3] first included the molecular signature in the diagnosis in glioma, and the 2021 WHO classification [4] clarified the value of the molecular phenotype again. For the last decades, many key molecular features have been recognized, such as isocitrate dehydrogenase (IDH) mutation, Chromosome 1p/19q deletion, O-6-Methylguanine-DNA Methyltransferase (MGMT) promoter methylation, Epidermal Growth Factor Receptor (EGFR) mutations, Telomerase Reverse Transcriptase (TERT) promoter mutations, etc. [2,5]. However, the roles of long noncoding RNA (lncRNAs) in glioma have not been fully explored.

LncRNAs are RNA transcripts longer than 200 nucleotides that do not encode any protein because of the lack of an open reading frame (ORF) [6]. Initially, lncRNAs were regarded as transcriptional noise because of the low sequence conservation. Subsequently, some researchers have pointed out that the lncRNAs might have various functions in biological processes by interacting with the related biomolecular [5]. In the last decades, an increasing number of studies have focused on the mechanisms of lncRNAs. Some lncRNAs with known functions have been discovered, such as X Inactive Specific Transcript (Xist) and human accelerated region 1 (HAR1), and the potential of application in cancer treatment is now receiving more attention in medical research. In glioma, HOXA Cluster Antisense RNA 2 (HOXA-AS2) was upregulated and promoted biological behaviors of malignant glioma and vasculogenic mimicry (VM) formation via the miR-373/EGFR axis [7]. Brain Cytoplasmic RNA 1 (BCYRN1) binding with miR-619-5p can regulate CUE Domain Containing 2 (CUEDC2) expression and the PTEN/AKT/p21 pathway to inhibit glioma tumorigenesis [8]. Small Nucleolar RNA Host Gene 25 (SNHG25) can promote the progression of glioma cells by activating MAPK Signaling [9]. Therefore, lncRNAs play an important role in the development of glioma. As a member of them, however, the biological function of TRHDE-AS1 is unknown. At present, the biological function of Thyrotropin Releasing Hormone Degrading Enzyme Antisense RNA 1 (TRHDE-AS1) is poorly understood. It is found on the q21.1 of chromosome 12.

In this study, we explored the expression level and prognostic value of TRHDE-AS1 in glioma, and we tried to study the biological mechanisms via bio-informative algorithms. We found that TRHDE-AS1 is a potential prognostic biomarker in glioma.

## 2. Materials and Methods

### 2.1. Data

The data was downloaded from the Cancer Genome Atlas Program (TCGA) database (https://xenabrowser.net/, accessed on 7 February 2023). The Brain Lower Grade Glioma (LGG) and Glioblastoma Multiforme (GBM) cohorts were chosen, which contain the RNAseq gene expression files, phenotype files, and clinical survival files. The mutation and molecular subtype files were downloaded by the “TCGAbiolink” package. By using the R language, we integrated those different data types into one matrix file. We note that the new version of the guidelines for the clinical diagnosis and treatment of glioma has made new standards for the classification of glioma, but the data collected in this paper are from an earlier time point and there is no up-to-date classification data, so we still use the old version of the classification standard.

### 2.2. Differential Expression Analysis

The samples were separated into a low expression group and a high expression group based on the expression level of TRHDE-AS1. The “limma” package was utilized to select the differential expression genes. The filtering criteria followed as adj.P.val (adjust *p* value) <0.05, and the absolute value of log2FC (fold change) >1.

### 2.3. Functional Analysis

Based on the expression of TRHDE-AS1, we separate the samples into high-risk and low-risk groups. The “clusterProfiler” package in R was utilized to analyze over-representation analyses of Gene Ontology (GO), false discovery rate (FDR) < 0.05, *p* value < 0.01, and Kyoto Encyclopedia of Genes and Genomes (KEGG), false discovery rate (FDR) < 0.01, *p* value < 0.01 between the high-risk and low-risk groups. Furthermore, function or pathway terms with an adjusted *p*-value < 0.05 and false discovery rate (FDR) < 0.25 were considered statistically significantly enriched.

### 2.4. GEPIA Database Analysis

The Gene Expression Profiling Interactive Analysis (GEPIA) database (http://gepia.cancer-pku.cn/, accessed on 7 February 2023) was used to compare the expression of TRHDE-AS1 between tumor and normal tissue in pan-cancer, which was applied by the “Gene Expression Profile” module. We choose to separate the patients into two groups according to the median expression of TRHDE-AS1, and to plot the Kaplan-Meier survival curve.

### 2.5. IntOgen Database Analysis

IntOgen database contains the most cancer-driver genes. We downloaded the driver genes list, and filtered the glioma-associated driver genes. As a result, in the sum of 568 driver genes, there were 58 genes left related to glioma, and these driver genes were extracted for the following analysis.

### 2.6. Immune Infiltration Analysis

CIBERSORTx (https://cibersortx.stanford.edu/, accessed on 7 February 2023) can calculate the fraction of immune infiltration cells by the gene expression data, which is based on a deconvolution algorithm. CIBERSORTx is an updated version of CIBERSORT.

### 2.7. Statistical Analysis

The Kaplan–Meier survival curve was used to evaluate the prognostic value of TRHDE-AS1 by performing log-rank tests to identify the significance of the difference between the survival curves. Differences between groups were compared using the Wilcoxon rank-sum test or the Student’s *t*-test, as appropriate. Correlations were determined using Pearson or Spearman correlation tests, as appropriate. All statistical *p* values were two-sided, and *p* < 0.05 was considered statistically significant. All analyses were performed with R software (version 4.2.1).

## 3. Results

### 3.1. Low Expression and Prognostics of TRHDE-AS1 in Pan-Cancer

Figure 1 shows the location of TRHDE-AS1 in the genome and its nearby genes. It is located in the chromosome 12 long arm q21.1 extents, the surrounding adjacent genes including TBC1 Domain Family Member 15 (TBC1D15), MRS2 Pseudogene 2 (MRS2P2), LOC112163631, Thyrotropin Releasing Hormone Degrading Enzyme (TRHDE), Tryptophan Hydroxylase 2 (TPH2), H3 Histone Pseudogene 35 (H3P35), and Coiled-Coil-Helix-Coiled-Coil-Helix Domain Containing 3 Pseudogene 2 (CHCHD3P2). As is shown in Figure 2a, TRHDE-AS1 is under-expressed in many cancers, which are Breast Invasive Carcinoma (BRCA), Cervical Squamous Cell Carcinoma and Endocervical Adenocarcinoma (CESC), LGG, GBM, Lung Adenocarcinoma (LUAD), Lung Squamous Cell Carcinoma (LUSC), Ovarian Serous Cystadenocarcinoma (OV), Pancreatic Adenocarcinoma (PAAD), Prostate Adenocarcinoma (PRAD), Testicular Germ Cell Tumors (TGCT), Uterine Corpus Endometrial Carcinoma (UCEC), and Uterine Carcinosarcoma (UCS). By plotting the KM survival curve (Figure 2b–g), it can be observed that in Esophageal Carcinoma (ESCA), GBM, LGG, Kidney Renal Clear Cell Carcinoma (KIRC), Kidney Renal Papillary Cell Carcinoma (KIRP), and Stomach Adenocarcinoma (STAD), TRHDE-AS1 has prognostic value. In KIRP, LGG, STAD, and KIRC, especially, it is statistically significant. However, it is not statistically significant in GBM, and there is a trend of a better prognostic in patients with a higher expression level of TRHDE-AS1, which is the same trend shown by LGG. Therefore, we believe that glioma TRHDE-AS1 does have prognostic value.

### 3.2. The Difference in Clinical Characteristics between High and Low Expression of TRHDE-AS1

As Figure 3 shows, we compared the expression level of TRHDE-AS1 in different molecular subtypes. According to the new WHO Classification of tumors of the central nervous system [4], adult-type diffuse gliomas are classified into three subtypes (Astrocytoma, IDH-mutant, Oligodendroglioma, IDH-mutant, and 1p/19q-deleted, and Glioblastoma, IDH-wildtype). Among the different subtypes, the expression level of TRHDE-AS1 shows a statistical difference. Beyond the new classification, it also suggests that among different forms of histology, IDH mutant, 1p/19q codeletion, and grade, there is a differential expression of TRHDE-AS1 and it is statistically significant.

### 3.3. Co-Expressing Genes of TRHDE-AS1 in Glioma

Pearson’s correlation analysis has recognized the co-expressing genes (Figure 4). As a result, in LGG, we found that Thyrotropin Releasing Hormone Degrading Enzyme (TRHDE), Transmembrane Protein 132D (TMEM132D), BASP1 Antisense RNA 1 (BASP1-AS1), HECT, C2 And WW Domain Containing E3 Ubiquitin Protein Ligase 1 (HECW1), Potassium Voltage-Gated Channel Subfamily C Member 1 (KCNC1), Contactin 4 (CNTN4), Potassium Sodium-Activated Channel Subfamily T Member 1 (KCNT1), WSC Domain Containing 2 (WSCD2), Cyclin And CBS Domain Divalent Metal Cation Transport Mediator 1 (CNNM1), and Regulating Synaptic Membrane Exocytosis 2 (RIMS2) were all positively related to the TRHDE-AS1. NECAP Endocytosis Associated 2 (NECAP2), SUZ RNA Binding Domain Containing 1 (SZRD1), Adenylate Kinase 2 (AK2), Ras Homolog Family Member C (RHOC), Solute Carrier Family 39 Member 1 (SLC39A1), Fc γ Receptor and Transporter (FCGRT), SPG21 Abhydrolase Domain Containing, Maspardin (SPG21), G Protein Subunit γ 5 (GNG5), Secretory Carrier Membrane Protein 2 (SCAMP2), G Protein Subunit α I3 (GNAI3), meanwhile, were all negatively related to TRHDE-AS1. In GBM, the co-expression analysis showed that the TRHDE, Calcium Homeostasis Modulator 1 (CALHM1), Ryanodine Receptor 2 (RYR2), LOC105376654, melanin concentrating hormone receptor 2 (MCHR2), fatty acid desaturase 6 (FADS6), ANKRD34C antisense RNA 1 (ANKRD34C-AS1), serine/threonine/tyrosine kinase 1 (STYK1), 5-hydroxytryptamine receptor 2C (HTR2C), NACHT and WD repeat domain containing 2 (NWD2) were all positively related to TRHDE-AS1, and the expression level of CKLF such as MARVEL Transmembrane Domain Containing 3 (CMTM3), Vimentin (VIM), aph-1 homolog A, γ-secretase subunit (APH1A), solute carrier family 39 member 1 (SLC39A1), profilin 1 (PFN1), SUZ RNA binding domain containing 1 (SZRD1), G protein subunit γ 5 (GNG5), NECAP endocytosis associated 2 (NECAP2), ribosomal protein lateral stalk subunit P0 pseudogene 6 (RPLP0P6), and TGFB induced factor homeobox 1 (TGIF1) were negatively related to TRHDE-AS1. We have noticed that LGG and GBM share the same positively co-expressing gene of TRHDE and negatively co-expressing genes of NECAP2, SZRD1, SLC39A1, and GNG5.

### 3.4. Functional Analysis of TRHDE-AS1 in Glioma

To explore the mechanism changes derived by TRHDE-AS1 in glioma, we applied the enrichment analysis. The GO and KEGG ORA analysis results of LGG and GBM are shown in Figure 5. As is shown in the figure, we can find that the TRHDE-AS1 may affect the functions of the membrane and synapse in both LGG and GBM. In some signal pathways, TRHDE-AS1 could affect the calcium signaling pathway, cAMP signaling pathway, GABAergic synapse, etc. These suggest the possibility of influence caused by TRHDE-AS1 on the neurocyte. This was consistent with the results of co-expression analysis, in which multiple genes were involved in the regulation of membrane function, such as KCNC1, GNG5, SCAMP2, GNAI3, HTR2C, CMTM3, APH1A, and PFN1.

### 3.5. Association between TRHDE-AS1 Expression and Cancer Driver Genes

The intOgen database contains the fullest list of cancer driver genes. After downloading the driver genes list and filtering, 58 driver genes of glioma were left. The Pearson correlation was as shown in Figure 6. Among the 58 genes, there were 37 genes associated with the expression levels of *TRHDE-AS1* in low grade glioma, and 22 genes associated with TRHDE-AS1 in high grade glioma. Among these genes, 15 genes are shared in both, which are AT-rich interaction domain 1B (ARID1B), BCL11 transcription factor B (BCL11B), B-Raf proto-oncogene, serine/threonine kinase (BRAF), calcium voltage-gated channel subunit alpha1 D (CACNA1D), cyclin dependent kinase inhibitor 2C (CDKN2C), Isocitrate Dehydrogenase (NADP (+)) 1 (IDH1), Lysine Methyltransferase 2D (KMT2D), LDL Receptor Related Protein 1B (LRP1B), Mechanistic Target Of Rapamycin Kinase (MTOR), NRAS proto-oncogene, GTPase (NRAS), protection of telomeres 1 (POT1), RB transcriptional corepressor 1 (RB1), RUNX1 Partner Transcriptional Co-Repressor 1 (RUNX1T1), tet methylcytosine dioxygenase 1 (TET1), and tumor protein p53 (TP53). As is plotted in Figure 5a–f, the correlation between TRHDE-AS1 and BRAF, IDH1, and TP53 shows a positive relationship of BRAF and TRHDE-AS1 in both LGG and GBM and a negative relationship of IDH1, TP53, and TRHDE-AS1 in LGG and GBM.

### 3.6. TRHDE-AS1 Associated Mutation Profile of Glioma

From the 58 driver genes in low grade glioma, the top 10 mutated genes were displayed and annotated for the risk group in Figure 7a. The IDH1, TP53, ATRX chromatin remodeler (ATRX), capicua transcriptional repressor (CIC), and other genes have a very high mutation frequency, IDH1 with 75%, TP53 with 46%, ATRX with 36%, and CIC with 22%. In both low-risk patients and high-risk patients, the IDH1, TP53, and ATRX have a high mutation frequency, but CIC and FUBP1 prefer to have a higher mutation frequency in low-risk patients and EGFR and PTEN have a higher mutation frequency in high-risk patients.

The mutation landscape of GBM was shown in Figure 7b. This figure shows the mutated genes are quite different from LGG. Mutations of phosphatase and tensin homolog (PTEN) and TP53 are the most frequent in high-risk and low-risk groups, PTEN with 14% and TP53 with 14%. Mutation of EGFR in high-risk is more frequent than low-risk groups. The TP53 shows a higher trend of different types of mutation, while the most common mutations are missense mutations.

### 3.7. Association between TRHDE-AS1 Expression and Immune Infiltration Levels

As Figure 8 shows, the correlation analysis between the immune cell fraction and TRHDE-AS1 expression level demonstrates that the memory B cells, CD8 T cells, CD4 naïve T cells, T follicular helper cells, activated NK cells, activated mast cells, eosinophils, and Neutrophils are positively correlated with TRHDE-AS1 expression, and T regulatory cells, M2 macrophages are negatively correlated with TRHDE-AS1. These correlations are statistically significant. This explains why TRHDE-AS1 may act as a protective factor. This is because the expression level of this gene to some extent represents a negative correlation between the glioma microenvironment and immune cells, such as T regulatory cells, M2 macrophages that promote tumor progression.

## 4. Discussion

In this study, we explored the prognostic value of TRHDE-AS1 in glioma and the association between it and genomic features, and we studied the potential function of TRHDE-AS1 in the signal pathway and immune microenvironment by bioinformatic algorithms. In pan-cancer analysis, the TRHDE-AS1 was demonstrated to have prognostic value in many cancers, including glioma. When applied to glioma, TRHDE-AS1 is a protective factor. With a deeper exploration, we found that the expression level of TRHDE-AS1 was correlated to clinical features such as grade, histology, IDH mutation, etc. In previous research, the prognosis of glioblastoma is the worst, with less than 14 months of median survival while the prognosis of oligodendroglioma is the one of the best, which is the same as our results. We found that the expression level of TRHDE-AS1 in oligodendroglioma is higher than in glioblastoma and the expression in WHO II is higher than in WHO IV with the higher grade, meaning worse prognosis. The expression level of TRHDE-AS1 in IDH mutation, 1p/19q chromosome codeletion is also higher than IDH wild type and 1p/19q non-codeletion, and this result accords with existing research, suggesting that IDH mutation and 1p/19q codeletion are the favorable factors [10,11,12].

As the most common malignant primary brain tumors, glioma is different from solid tumors. Because of the unexpected facts of treatment resistance in many clinical trials on glioma [13], it is still a challenge to improve the prognosis. Currently, more and more researchers have suggested novel treatments, such as IDH inhibitors [14,15,16], EZH2 inhibitors [17], HDAC inhibitors [18], DNMT inhibitors [19,20], etc. Although most treatments have failed to prolong the patient’s survival, there are still a few that have attracted the attention of scientists, and there are an increasing number of studies focusing on the mechanisms causing the difference in prognosis. Meanwhile, computing methods have joined the field with the extraordinary development of computer science. With the development of computer science and molecular biology, high-throughput sequencing technology is widely used in medical research. In glioma, an increasing number of molecular features are found, which play a very important guiding role in the diagnosis and clinical treatment of glioma. So far, these biomarkers are mostly the protein-coding genes, whose biological functions have been well understood. However, the protein-coding genes only occupy a small fraction of the human genome, and most parts of the genome are non-coding. Many works have demonstrated that lncRNAs play an important role in cancer development, such as PVT1 [21] and CCAT2 [22,23] in colorectal cancer, PCAT-1 [24,25] in prostate cancer, PTCSC3 [26] in thyroid cancer, HULC [27], GAS5 [28] in hepatocellular cancer, ANRIL [29] in various cancer, TERC [30] in oral cavity cancer, etc. These lncRNAs are widely used in cancer diagnosis and monitoring, prognosis, and therapeutic responsiveness prediction. The main function of LncRNA is to regulate the function of RNA. Therefore, in this study, we found that TRHDE-AS1 is supposed to mediate the key genes during the process of origin and the development of glioma. However, the function of lncRNA TRHDE-AS1 still needs to be studied more deeply. Unfortunately, there are few studies in the field of glioma. In breast cancer, Hu [31] found the ability of astragaloside IV through promoting the expression of TRHDE-AS1 to inhibit cancer cell growth. Wei [21] proposed the ability of TRHDE-AS1 as a “miRNA sponge” to inhibit the proliferation and apoptosis in hypertrophic scar fibroblasts (HSFs) through TRHDE-AS1/miR-181a-5p/PTEN pathway. Zhuan [22] found that TRHDE-AS1 can overexpress to inhibit cell proliferation and invasion in lung cancer through the miRNA-103/KLF4 axis. However, this research has proved unable to explain the exact mechanism of TRHDE-AS1 in glioma. As an exploratory study, our research suggests that TRHDE-AS1 can possibly regulate the membrane and signal pathways, which need to be verified in the experiment. In co-expressing analysis, we found that LGG and GBM share the same positively co-expressing gene of TRHDE and negatively co-expressing genes of NECAP2, SZRD1, SLC39A1, and GNG5. Adaptin ear-binding coat-associated protein 2 (NECAP2) belongs to the family of proteins encoding adaptin-ear-binding coat-associated proteins. In glioma, the overexpression of NECAP2 has a remarkably higher risk of developing malignant behavior and a worse prognosis [32]. SZRD1 is considered to be a gene associated with the prognosis of glioma. Deng and Geng et al. have found that multiple lncRNAs participate in the regulation of the occurrence and development of glioma by targeting the miR-128-3p/SZRD1 and miR-638/SZRD1 axis. It has also been reported that SLC39A1 is highly expressed in glioma tissue, which can promote the proliferation of tumor cells, inhibit apoptosis, and affect the level of immune infiltration in the glioma microenvironment [33]. Zhang and Yang found that GNG5 was an unfavorable prognostic factor in glioma, and it can promote the proliferation and migration of glioma cells [34,35]. Through multiple bioinformation algorithms, they found that GNG5 expression was associated with WHO grade, age, histology, recurrence, necrosis, and microvascular proliferation. It should also be noted that the function of KCNC1, KCNT1, RIMS2, NECAP2, GNG5, SCAMP2, GNAI3 genes is also correlated with membrane function in low-grade gliomas. In addition, BASP1-AS1, RHOC and SLC39A1 have been reported to be associated with cancer. In high-grade gliomas, RYR2, MCHR2, FADS6, HTR2C, CMTM3, APH1A, and PFN1 genes are related to membrane function. In conclusion, a large part of related genes obtained from TRHDE-AS1 co-expression analysis are involved in the regulation of membrane function. This is consistent with the results of the subsequent functional analysis, so this result is also persuasive. Yang et al. found that KCNC1 may be involved in the regulation of ion channel-related functions, and added it as a member to the glioma prognostic model he constructed [36]. RIMS2 has also been reported to be associated with neuron-related functions and is specifically expressed in oligodendroglioma with 1p loss [37]. Xu et al. found that the function of BASP-AS1 is related to hypoxia, and is involved in the process of regulating the microenvironment of glioma through hypoxia-related signaling pathways, which affects the prognosis of glioma [38]. Zhang et al. found that RHOC promoted the tumor progression of glioma by interacting with P53, extracellular matrix receptor, and focal adhesion pathways [39]. At the same time, RHOC is also a member of RHOC/Cofilin signaling pathway, and several genes can be involved in glioma mechanism by regulating this signaling pathway [40,41]. Li et al. found that CMTM3 was significantly associated with a variety of immune cells in the immune microenvironment, such as macrophages and dendritic cells in high-grade gliomas, and was also associated with prognosis. High expression of CMTM3 was associated with a poorer prognosis [42]. Fan et al. reported that Pfn-1 phosphorylation is involved in the regulation of endothelial vasculin expression and promotes abnormal vascularization and GBM development through a hypoxy-independent HIF-1α induction mechanism [43]. These co-expressed genes constitute a huge interacting regulatory network involved in the regulation of glioma related pathways and further affect the progression of glioma. Therefore, the involvement of TRHDE-AS1 in the progression of glioma is credible and needs to be confirmed by more experimental means.

We also explored the association between TRHDE-AS1 and cancer-diver genes. The studies on the genetic mechanisms of cancer have proposed the concept of the driver gene, suppressor gene, and oncogenic gene. In 1979, in order to find the protein interacting with the viral SV40 T antigen, TP53 was found by Kress, Lane, Linzer, et al. [23,24,25], and identified as an oncogenic gene subsequently in the 1980s [44,45,46,47]. However, with deeper exploration, scientists found that TP53 was a suppressor gene [30,44,45,46,48,49,50]. Currently, with the development of high-throughput sequencing technology, an increasing number of studies have found many key genes, functioning as described above, such as EGFR, ALK Receptor Tyrosine Kinase (ALK), KRAS Proto-Oncogene, GTPase (KRAS), MET Proto-Oncogene, Receptor Tyrosine Kinase (MET), B-Raf Proto-Oncogene, Serine/Threonine Kinase (BRAF), Ret Proto-Oncogene (RET), etc. [47]. With the development of genetics in glioma, molecular characteristics have been incorporated into the new diagnostic system [4]. In this study, we found that TRHDE-AS1 is correlated to many cancer driver genes of glioma, especially BRAF, IDH1, and TP53, which means the TRHDE-AS1 does participate in the biological processes related to glioma. This regulation could be a complex network. We could see that the correlation between TRHDE-AS1 and these three genes was the same in low-grade gliomas and high-grade gliomas, which also indicated that TRHDE-AS1 had similar anti-cancer effects between them. TRHDE-AS1 was negatively correlated with the expression levels of several oncogenes and positively correlated with the expression levels of several tumor suppressor genes. This is consistent with the results of its own survival analysis. The mutual testimony of these two factors once again indicated the unique role of TRHDE-AS1 in glioma.

There are still some limitations to our research. First of all, our work is implemented based on the bioinformatic algorithm, and almost all of the conclusions are obtained through statistics and probability calculation. These accounting methods do not take into account relevant biological mechanisms, but are still instructive. Second, our study is retrospective, and the timeliness of the data is poor, so the data quality cannot be improved in real time based on the research content. Third, the results of the computational methods still need to be verified in the laboratory. We believe that the results of our rigorous algorithmic exploration and analysis of glioma gene expression data will have important guiding significance for future mechanism studies. Through experimental methods, the biological principle of lncRNA TRHDE-AS1’s involvement in the occurrence and development of glioma will be found.

## 5. Conclusions

In conclusion, we explored the prognostic value based on the expression level of TRHDE-AS1 by lasso regression to predict clinical outcomes in glioma patients, which suggested that this lncRNA had great potential to guide clinical prognosis prediction and decision making for treatment in the future.

## Figures and Tables

**Figure 1 genes-14-01052-f001:**
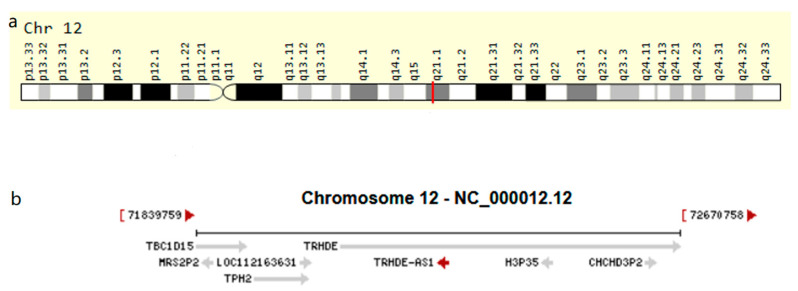
A schematic of TRHDE-AS1. (**a**) the location of TRHDE-AS1 in genome; (**b**) the length of TRHDE-AS1 and its nearby genes.

**Figure 2 genes-14-01052-f002:**
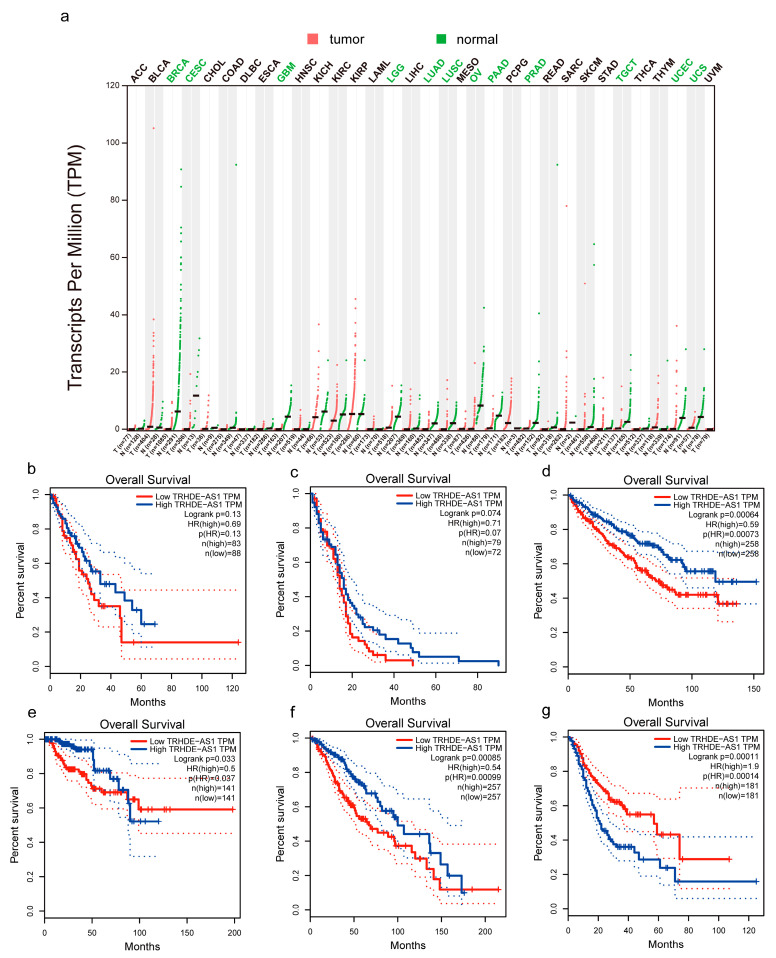
The expression level and prognostic value of TRHDE-AS1 in pan cancer. (**a**) The expression of TRHDE-AS1 in pan cancer; (**b**–**g**) the prognostic value of TRHDE-AS1 in ESCA, GBM, KIRC, KIRP, LGG, and STAD.

**Figure 3 genes-14-01052-f003:**
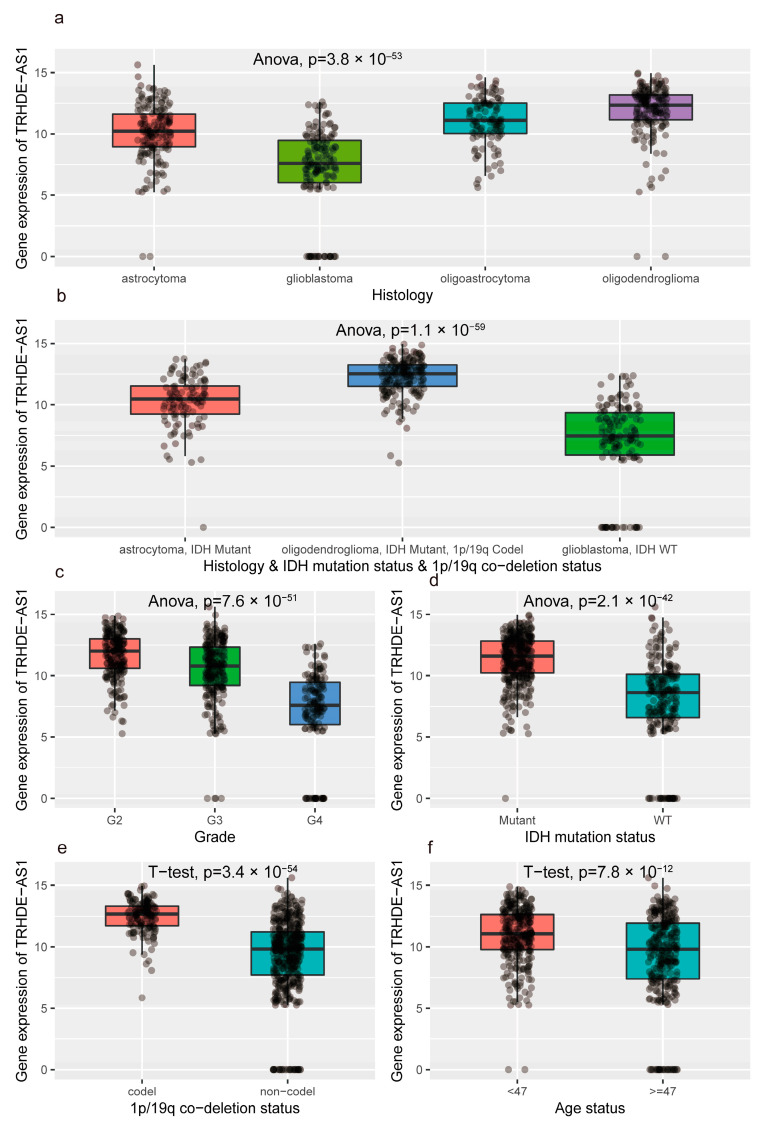
Comparison of TRHDE-AS1 expression and molecular subtypes: (**a**) TRHDE-AS1 expression based on histology; (**b**) TRHDE-AS1 expression based on Histology, IDH mutation status and 1p/19q co-deletion status; (**c**) TRHDE-AS1 expression based on grade; (**d**) TRHDE-AS1 expression based on IDH mutation status; (**e**) TRHDE-AS1 expression based on 1p/19q co-deletion status; and (**f**) TRHDE-AS1 expression based on age status.

**Figure 4 genes-14-01052-f004:**
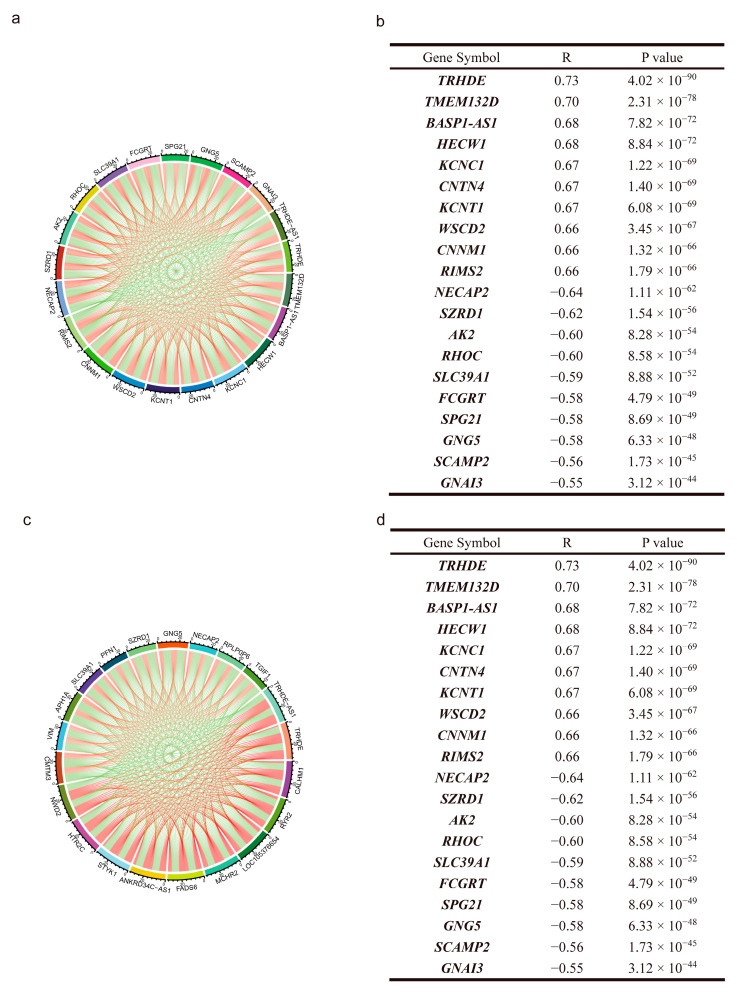
Co-expression genes of *TRHDE-AS1* in glioma: (**a**,**b**), co-expression of *TRHDE-AS1* in LGG (red means positive correlation, green means negative correlation); (**c**,**d**), co-expression of *TRHDE-AS1* in GBM (red means positive correlation, green means negative correlation). R is the Pearson correlation coefficient.

**Figure 5 genes-14-01052-f005:**
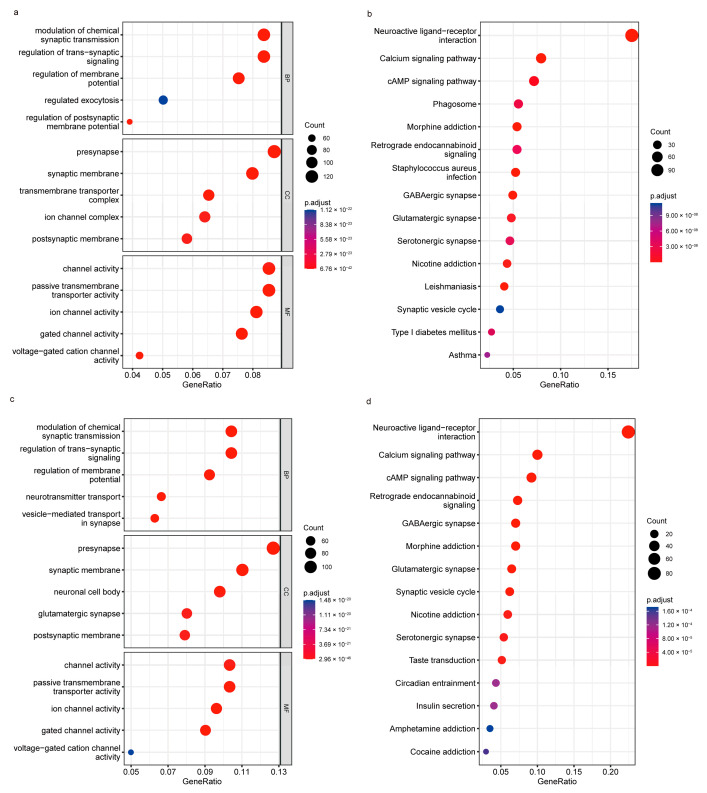
Functional analysis of TRHDE-AS1: (**a**) GO over-representation analysis of LGG; (**b**) KEGG over-representation analysis of LGG; (**c**) GO over-representation analysis of GBM; and (**d**) KEGG over-representation analysis of GBM. Gene Ratio is the percentage of total DEGs in the given term.

**Figure 6 genes-14-01052-f006:**
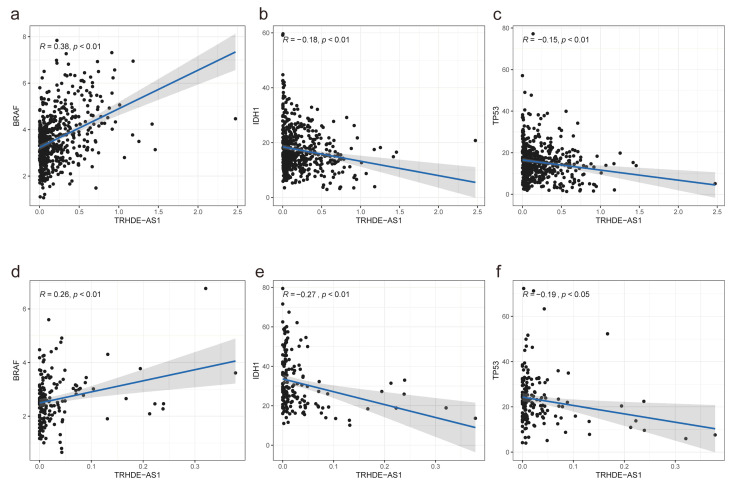
Association between TRHDE-AS1 and cancer driver genes: (**a**) the correlation between BRAF expression and TRHDE-AS1 in LGG; (**b**) the correlation between *IDH1* expression and TRHDE-AS1 in LGG; (**c**) the correlation between TP53 expression and TRHDE-AS1 in LGG; (**d**) the correlation between BRAF expression and TRHDE-AS1 in GBM; (**e**) the correlation between IDH1 expression and TRHDE-AS1 in GBM; (**f**) the correlation between TP53 expression and TRHDE-AS1 in GBM.

**Figure 7 genes-14-01052-f007:**
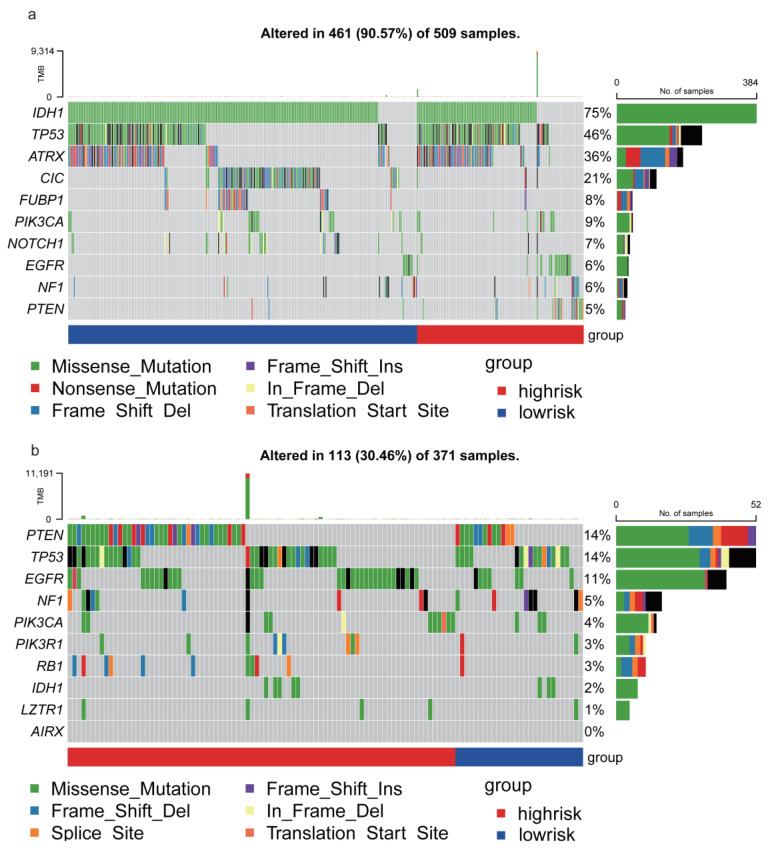
TRHDE-AS1 associated mutation landscape in glioma (**a**) the oncoplot of LGG; (**b**) the oncoplot of GBM.

**Figure 8 genes-14-01052-f008:**
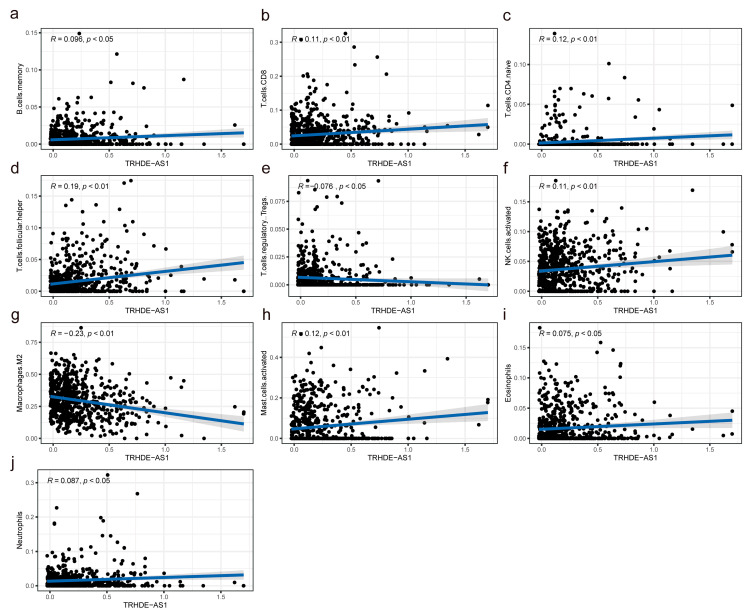
Correlation of TRHDE-AS1 and tumor immune components: (**a**–**j**) correlation between TRHDE-AS1 expression and memory B cells, CD8 T cells, CD4 T cells, follicular helper T cells, regulatory T cells, NK cells, M2 macrophages, mast cells, eosinophils, and neutrophils.

## Data Availability

The glioma datasets generated and analysed during the current study are available in the TCGA project (http://xenabrowser.net/, accessed on 7 February 2023).

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
