# Peer review of "lncRNA TRHDE-AS1 Correlated with Genomic Landscape and Clinical Outcome in Glioma"

_genes, 2023, doi:10.3390/genes14051052_

Round 1

Reviewer 1 Report

Xie et al carried out various bioinformatic analyses to derive correlations between lncRNA TRHDE-AS1 expression and various parameters of glioma.  Although various correlations are shown, a lack of insight about how these correlations can be utilized remains a concern. Specific comments are as follows:

1.       What is TRHDE-AS1?  The full name of the lncRNA must be provided.  Also, full name of all genes under sections 3.3, 3.5 and everywhere else (for example, in the legend to figure 1) need to be provided.  It is not a good idea to expect that a reader or a reviewer would know full names of all the abbreviated genes.

2.       A schematic indicating location of TRHDE-AS1 in the genome, and nearby genes, must be provided.

3.       In figure1, indicate what black and green names mean? Also, what do the red and green dots mean?

4.       The weakness in English is spread throughout the manuscript that needs to be resolved. An example: the legend to Figure 2 should be: Comparison of TRHDE-AS1 expression and molecular subtypes (delete ‘among’). a, TRHDE-AS1 expression based on histology. (not ‘in’). The rest of the legend should be changed this way as well.

5.       Figure 3: Full gene names, please.  Also, define ‘R’.  Explain what a high ‘R’ value and a low ‘R’ value would imply.

6.       Figure 4: Explain what ‘GeneRatio’ mean. Also, it seems the plots in b and d are almost identical – please explain this.

7.       Section 3.5: TRHDE-AS1 has been shown to be positively associated with some genes and negatively associated with another set of genes.  I feel that with this kind of analysis, any gene probably will be positively associated with a set of genes, and negatively associated with a different set of genes.  So, unless such associations are explained well, its value remains questionable.

8.       Legend to Figure 5: correct ‘d’ and ‘g’

9.       Discussion: Most or all the paragraphs 2,3,4, and 5 (Lines 221- 269) are appropriate for the Introduction not Discussion.

10.   In the Conclusion, it is implied that measuring the TRHDE-AS1expression level will have predictive value.  To arrive at such a conclusion, I feel the authors, at minimum, should have worked with two groups of samples: a working/practice group and a test/confirmation group.

Reviewer 2 Report

The authors of, "lncRNA TRHDE-AS1 correlated with genomic landscape and clinical outcome in glioma" have presented a bioinformatic study of public gene expression datasets of glioma. They have identified TRHDE-AS1 as a potential gene of interest with respect to predicting patient outcomes at the time of diagnosis, a laudable goal with the limited therapies available for treatment. The major limitation of the value of the study was identified by the authors in the discussion, "These accounting methods do not take into account relevant biological mechanisms..." Despite the fact that this study was completely in silico, it would still be useful for the authors to describe the function of the TRHDE gene product, and discussion of the consequences should the TRHDE-AS1 inhibit the gene product which is a reasonable potential outcome, and relate that to the context of the cancer. This sort of discussion would add to the potential value of the manuscript. 

Also there are some additional points that detract from the manuscript. Overall the figures are too low resolution and the text is difficult to read. Figure 6 is completely illegible.  What does the green vs the red signify in Figure 1A? The correlations of Figures 5-7 may have achieved statistical significance, but there should be some caution when interpreting this as the vast majority of the datapoints are clumped with just a few outliers driving the differences. Fig 2A includes the designation oligoastro which was removed from the classification of glioma in the 2016 WHO. Although these data are from the time when that designation was valid, this should be described in the manuscript. 

Additional points are that references 1 and 2 are the same, and the manuscript does not contain an abbreviation list for the cancer types, nor does it list the gene names, only the gene symbols, so that it is challenging for readers to make their own judgements of context of the data.

Overall, enthusiasm for this manuscript in its current form is very low. 

Round 2

Reviewer 1 Report

The revised version of the manuscript by Xie a et al appears to be much improved compared to the initial version.  My only comment currently is:

Change the legend to figure 3 as: TRHDE-AS1 expression based on histology, etc.

Reviewer 2 Report

The authors' revised manuscript is greatly improved. The figures are all legible providing a greater clarity of the data they are presenting. The context of the findings have been clarified with the revised discussion.  
